# AGENTIF: Benchmarking Instruction Following of Large Language Models in Agentic Scenarios

**Yunjia Qi**[1][*]  **Hao Peng**[1][*]  **Xiaozhi Wang**[2]  **Amy Xin**[1]  **Youfeng Liu**[4]
**Bin Xu**[1,3][†]  **Lei Hou**[1,3]  **Juanzi Li**[1,3]

[1]Department of Computer Science and Technology, BNRist;
[2]Shenzhen International Graduate School;
[3]KIRC, Institute for Artificial Intelligence,
Tsinghua University
[4]Zhipu AI

{qyj23, peng-h24}@mails.tsinghua.edu.cn

## Abstract

Large Language Models (LLMs) have demonstrated advanced capabilities in real-world agentic applications. Growing research efforts aim to develop LLM-based agents to address practical demands, introducing a new challenge: agentic scenarios often involve lengthy instructions with complex constraints, such as extended system prompts and detailed tool specifications. While adherence to such instructions is crucial for agentic applications, whether LLMs can reliably follow them remains underexplored. In this paper, we introduce AGENTIF, the first benchmark for systematically evaluating LLM instruction following ability in agentic scenarios. AGENTIF features three key characteristics: (1) Realistic, constructed from 50 real-world agentic applications. (2) Long, averaging $1{,}723$ words with a maximum of $15{,}630$ words. (3) Complex, averaging 11.9 constraints per instruction, covering diverse constraint types, such as tool specifications and condition constraints. To construct AGENTIF, we collect 707 human-annotated instructions across 50 agentic tasks from industrial application agents and open-source agentic systems. For each instruction, we annotate the associated constraints and corresponding evaluation metrics, including code-based evaluation, LLM-based evaluation, and hybrid code-LLM evaluation. We use AGENTIF to systematically evaluate existing advanced LLMs. We observe that current models generally perform poorly, especially in handling complex constraint structures and tool specifications. We further conduct error analysis and analytical experiments on instruction length and meta constraints, providing some findings about the failure modes of existing LLMs. We have released the code[3] and data[4] to facilitate future research.

## 1 Introduction

Large language models (LLMs) have demonstrated strong capabilities in real-world agentic applications [19]. Growing studies focus on developing LLM-based agents to address practical demands, such as Web Agents [7, 8], Education Agents [4], GUI Agents [31, 32], and PPT Agents [34]. While these agentic scenarios expand the application scope of LLMs, they also pose a new challenge:

---

[*]Equal contribution.

[†]Corresponding author: xubin@tsinghua.edu.cn

[3]https://github.com/THU-KEG/AgentIF

[4]https://huggingface.co/datasets/THU-KEG/AgentIF

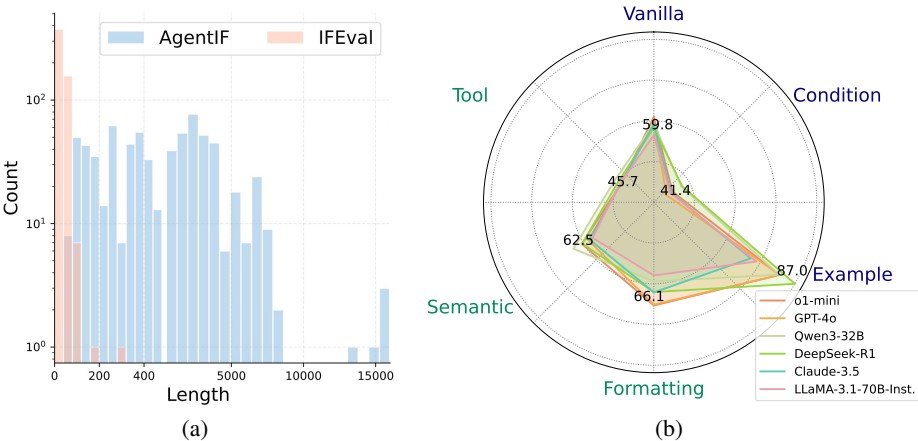

(a)              (b)

Figure 1: **(a)** The length distribution of instructions across AGENTIF (log-scale). **(b)** Success rates of several representative LLMs on different constraint dimensions (detailed descriptions are in § 3.2).

agentic tasks usually involve long and complex instructions, such as extended system prompts and detailed tool specifications. Correctly following such instructions is a prerequisite for solving these tasks and reflects a fundamental capability of LLMs.

However, whether LLMs can effectively follow instructions in real-world agentic scenarios remains underexplored. Previous work on benchmarking instruction following of LLMs has mainly focused on relatively short instructions, which are usually synthetically generated. For example, the widely used benchmark IFEval [36] is synthetically constructed with various constraint types, such as formatting, and has an average instruction length of only 45 words. Subsequent studies have expanded the instruction scope to include more constraint types [14, 23, 33, 29], system prompts [22], multi-turn conversations [10], and action constraints [16]. Nonetheless, the instructions in these benchmarks are typically short and usually synthetically generated, resulting in a gap from real-world agentic applications. Figure 2 illustrates an instruction from a real-world agentic scenario. We can observe that the instruction is long with complex structures and constraint types, such as condition constraints, example constraints, and tool specifications, posing novel and significant challenges for LLMs. As existing instruction-following benchmarks typically lack coverage of such agentic instructions, it is necessary to systematically evaluate LLMs' ability to handle them.

Considering the above concerns, we propose AGENTIF, the first benchmark to evaluate instruction following of LLMs in real-world agentic scenarios. Specifically, we first collect 50 agentic tasks from industrial applications and open-source agentic systems. For each task, we manually annotate around 20 user queries, each combined with the corresponding agentic system prompt to form an instruction. We then extract all the constraints from each instruction and investigate their types. As illustrated in Figure 2, we classify the constraints into three main types: (1) formatting constraints, which specify the structure or presentation of the output; (2) semantic constraints, which require semantic understanding to check, e.g., language style; (3) tool constraints, which involve adherence to tool specifications, e.g., the parameter format of a function. We also classify the representation types of these constraints into three types: (1) vanilla, which means the constraints is described directly in plain text; (2) condition, where constraints are triggered under certain conditions, e.g., "if the output exceeds 100 words, include the keyword *paper*"; (3) example, which is similar to in-context learning [3], where the model is expected to follow the structure shown in examples. Finally, AGENTIF includes 707 high-quality manually annotated instructions, with each instruction containing an average of 11.9 constraints. These real-world agentic instructions, with extended length (Figure 1(a)) and complex constraints, present significant challenges to existing LLMs. For the evaluation on AGENTIF, we annotate the evaluation method for each constraint, consisting of three paradigms: (1) code verification, which examines constraint satisfaction through Python code; (2) LLM verification, which assesses constraint satisfaction using large language models; (3) hybrid verification, which uses a combination of code and LLM verification. For example, when evaluating

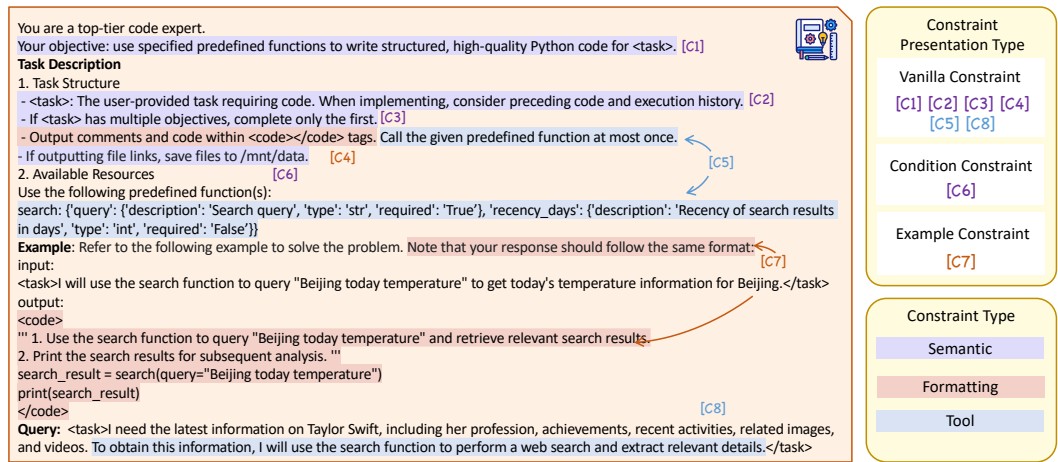

Figure 2: An example instruction of AGENTIF.

the constraint "The abstract should be no less than 100 words", the method first uses an LLM to extract the abstract section, followed by code verification to assess word count compliance.

We conduct comprehensive experiments to evaluate current advanced LLMs on AGENTIF. Specifically, we evaluate several representative thinking LLMs, non-thinking LLMs, and LLMs optimized for instruction-following. As shown in Figure 1(b), all LLMs perform poorly and even the best-performing model only follows fewer than 30% of the instructions perfectly. We further conduct error analysis and find that the most challenging condition and tool constraints introduce new challenges. We also find the performance declines as instruction length increases in AGENTIF. Additionally, we identify a novel category of constraints, meta constraints, which reflects underlying prioritization issues. In conclusion, advanced LLMs still struggle to follow real-world agentic instructions.

## 2 Related Work

Instruction following is a fundamental capability of LLMs, referring to following user instructions, including task completion and adherence to user requirements. The most widely used instruction-following benchmark is IFEval [37], which is the first to formalize the task as multi-constraint compliance, such as requirements on output length and format. For example, an instruction in IFEval "Write a long email, with at least 500 words. The email must include the keywords 'correlated' and 'experiencing'" includes constraints on output length, required keywords, which can be efficiently and accurately verified using Python code. Subsequent work has expanded instruction scope in several directions: (1) More constraint types [14, 23, 33, 29], which include constraints requiring semantic understanding (e.g., style) and adopt LLMs for evaluation. (2) Multilingual [33, 9, 10]. (3) Multi-turn [10, 9], which assesses instruction following in multi-turn dialogues, such as requiring the model to revise its response in the last round. (4) Code [30], which evaluates instruction-following capability in code generation. Notably, there are also two studies closely related to agentic scenarios. SysBench [22] evaluates compliance with system prompts, which are mined from realistic user logs. However, these prompts are typically short and lack tool usage, still leaving a gap from realistic agentic scenarios. AgentOrca [16] assesses adherence to operational constraints and routines of LLMs. They primarily focus on compliance with function invocation and do not involve the complex system prompts or constraint types typical of real-world agents. In conclusion, existing instruction-following benchmarks overlook the evaluation of instruction compliance in realistic agentic scenarios.

In realistic agentic scenarios, as shown in Figure 2, instructions are typically long with complex constraint types, structures, and tool specifications. As LLM-based agents are increasingly deployed across various domains [7, 8, 4, 31, 32, 34], accurate adherence to agentic instructions becomes essential. To address this need, we introduce AGENTIF, the first instruction-following benchmark for agentic scenarios. AGENTIF comprises data from real-world industrial applications and open-source agentic workflows, with comprehensive human annotations. Each instruction in AGENTIF contains $1,700$ tokens and $14$ constraints on average, which presents significant challenges to current LLMs.

# 3  AGENTIF

This section presents a detailed introduction to AGENTIF, including 4 parts: the constraint taxonomy (§ 3.1), the dataset construction process (§ 3.2), the statistics of AGENTIF (§ 3.3), and the evaluation protocol (§ 3.4). Figure 3 illustrates the dataset construction and evaluation workflow of AGENTIF.

## 3.1  Constraint Taxonomy

To comprehensively evaluate LLMs' ability to follow complex instructions in agentic settings, we investigate about 100 instructions from real-world scenarios and construct a constraint taxonomy, which classify constraints along two dimensions: constraint type, such as whether the constraint requires semantic understanding for verification, and constraint representation type, such as whether constraint is conditionally triggered. More details and examples are provided in Appendix C.

**Constraint Type**   The constraint type refers to the specific evaluation aspect, such as format or style. Following prior work [14, 29], we categorize constraints into two commonly used types and introduce a new type, that is tool constraints, specific to agentic scenarios. **Formatting constraints** specify the structure or presentation of the output. These include requirements about syntax (e.g., JSON or Markdown), layout (e.g., bullet points, tables, or paragraph length), symbol conventions (e.g., using backticks for filenames), and step-by-step formatting (e.g., explaining a principle in three steps). **Semantic constraints** focus on the semantic meaning and informativeness of the content, including requirements for depth, completeness (e.g., inclusion of keywords or references), and adherence to a specific style or tone. **Tool constraints** are newly introduced specifically for agentic scenarios, requiring the model to invoke tools according to given specifications, such as adhering to the correct parameter types, avoiding internet access, or restricting tool usage to a predefined set of functions.

**Constraint Presentation Type**   The constraint presentation type refers to how constraints are presented in text. We categorize this into three forms. **Vanilla constraints** are described explicitly in plain text (e.g., include the keyword *paper*). **Conditional constraints** are triggered only under specific conditions, which may be derived from the input (e.g., responding only if certain keywords appear) or from the model's own output behavior (e.g., applying markdown rules only when markdown is used). **Example constraints** are not explicitly stated but implied through few-shot examples, like in-context learning [3], requiring the model to infer and follow constraints from given output examples, which requires analogical reasoning and inductive capabilities.

## 3.2  Dataset Construction

As shown in Figure 3, AGENTIF is constructed through a semi-automated pipeline consisting of three main steps: agentic instruction collection, constraint extraction, constraint evaluation design. More details of the data construction process and human annotation are shown in Appendix D.

**Instruction Collection**   We focus on real-world agentic instructions, constructing our dataset from two sources: open-source agentic applications and industrial agent-based frameworks. We collect instructions based on two key principles: (1) **Realistic**: Each instruction should reflect practical, real-world agentic tasks. (2) **Complex**: Instructions should involve complex constraints, structures, and tool specifications that pose significant challenges for LLMs. Specifically, we first collect 40 agents from GitHub[5], including well-known agentic applications such as Cursor and Manus, along with 10 agents from industrial agentic workflows, which supports around 200 daily active users, handles approximately 300 requests per day, and has delivered a total of $120,000$ services. The 50 collected agents include only system prompt, which contain task specifications, goals, and tool descriptions, without any user queries, ensuring no risk of user data leakage. Next, we use GPT-4o to generate about 20 user queries for each agent based on their system prompts. We then apply heuristic rules and similarity-based filtering to remove redundant queries. Finally, we employ human annotators to rewrite each generated query, ensuring consistency with real user cases. As a result, we obtain 707 high-quality instructions with an average length of $1,723$ words.

---

[5]https://github.com/x1xhlol/system-prompts-and-models-of-ai-tools
https://github.com/Shubhamsaboo/awesome-llm-apps

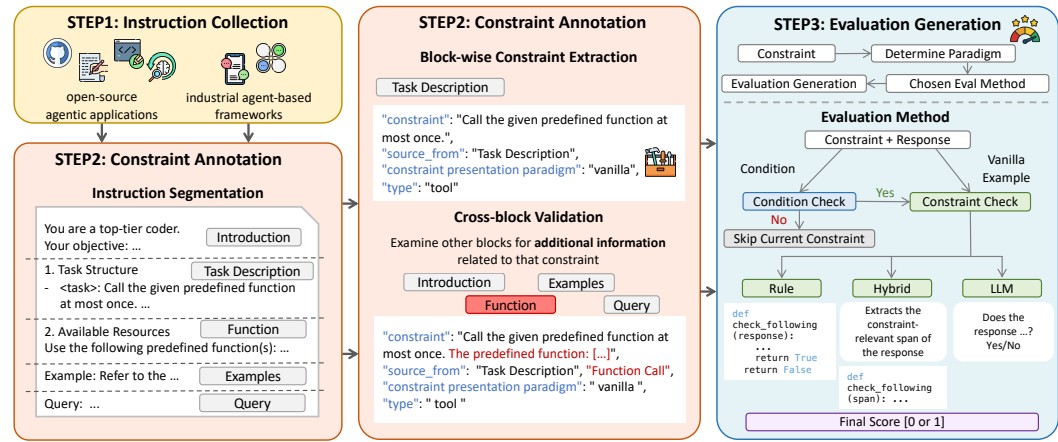

Figure 3: The data construction process and evaluation workflow of AGENTIF. The detailed descriptions of different constraint types are presented in § 3.1.

**Constraint Annotation**   As shown in Figure 3, we first use an LLM to automatically extract constraints from the instructions. As the original instructions are long with complex structure, directly extracting constraints poses a challenge for LLMs. To address this, we design a **block-wise** annotation method. Specifically, we segment each instruction into self-contained semantic blocks (e.g., task description, tool configuration, output specification), ensuring that the content is not truncated. We then use GPT-4o to extract relevant constraints from each block. Some constraints may span multiple blocks, for example in Figure 3, the complete tool specifications are distributed across multiple blocks. Therefore, we perform cross-block validation to add this information to ensure the completeness of each constraint. Finally, we employ human annotators to validate each constraint to verify its completeness and consistency with the original instruction. As a result, we obtain $8,415$ high-quality constraints in total, with an average of $11.9$ constraints per instruction.

**Evaluation Generation**   Finally, we annotate each constraint with its corresponding evaluation method. Following prior work [14, 23, 29], we adopt a hybrid evaluation framework that combines LLM-based and code-based evaluation. We adopt different evaluation methods for different types of constraints as described in § 3.1. Specifically, as shown in Figure 3, we define three evaluation modes based on constraint types: (1) **Code evaluation**, which is used for constraints that can be verified through simple and deterministic Python code (e.g., keyword presence, formatting patterns). (2) **LLM evaluation**, which is applied to open-ended or subjective constraints requiring semantic understanding. In these cases, we adopt an LLM for evaluation. (3) **Hybrid evaluation**, which is used for more complex cases, where the LLM first identifies relevant segments in the response (e.g., extracting JSON for tool calls), followed by code-based validation. Notably, for conditional constraints, we annotate the evaluation process to first check whether the condition is met before performing constraint evaluation. We use GPT-4o to determine the evaluation method for each constraint and to generate the corresponding evaluation script. We then manually review all annotations to ensure the correctness of the generated evaluation methods and revise them as needed.

## 3.3   Data Statistics

The statistics of AGENTIF and other related benchmarks are summarized in Table 1. Compared to existing instruction-following benchmarks, AGENTIF features three key characteristics: (1) **Realistic**. AGENTIF is derived from real-world agentic scenarios which reflects real use cases. (2) **Long**. Instructions in AGENTIF are significantly longer than those in prior benchmarks, with an average length of $1,723$ words. We also illustrate the length distribution of AGENTIF in Figure 1(a). We can observe that a substantial portion of instructions even exceeds $3,000$ words, posing a significant challenge for existing LLMs. (3) **Complex**. Each instruction contains an average of $11.9$ constraints, with a good coverage of various constraint types such as tool, condition, and example constraints. Further statistics about the dataset can be found in Appendix D.3.

| Benchmark | #Inst. | Len. | #Cons. | Data Resource | Constraint Type | | | Evaluation Method | |
|---|---|---|---|---|---|---|---|---|---|
| | | | | | Tool | Conditional | Example | Code-based | LLM-based |
| IFEval [37] | 541 | 36 | 1.5 | Synthetic | ✗ | ✗ | ✗ | ✓ | ✗ |
| FollowBench [14] | 820 | 253 | 3.0 | Synthetic | ✗ | ✗ | ✓ | ✓ | ✓ |
| InfoBench [23] | 500 | 38 | 4.5 | Synthetic | ✗ | ✗ | ✗ | ✗ | ✓ |
| SysBench [22] | 500 | 521 | 2.4 | Realistic | ✗ | ✗ | ✗ | ✗ | ✓ |
| ComplexBench [29] | 1,150 | 448 | 4.2 | Synthetic | ✗ | ✓ | ✗ | ✓ | ✓ |
| AgentOrca [16] | 663 | 1,144 | - | Synthetic | ✓ | ✓ | ✗ | ✓ | ✗ |
| Multi-IF [10] | 4,501 | 48 | 7.1 | Synthetic | ✗ | ✗ | ✗ | ✓ | ✗ |
| AGENTIF (ours) | 707 | 1,723 | 11.9 | Realistic | ✓ | ✓ | ✓ | ✓ | ✓ |

Table 1: The statistics of AGENTIF and previous instruction-following benchmarks. The statistics include dataset size (#Inst.), average instruction length (#Len.), average number of constraints per instruction (#Cons.), data resource, constraint types, and supported evaluation methods.

## 3.4 Evaluation Protocol

Figure 3 illustrates our evaluation methodology. The process first determines whether each constraint requires verification. Condition constraints that are not triggered are exclued from verification. We then adopt the annotated corresponding evaluation method. For LLM-based and hybrid evaluations, we employ `gpt-4o-2024-11-20`. To validate the reliability of this automated evaluator, we randomly sample 200 responses and compare its judgments with human annotations, with a $94\%$ agreement, supporting the use of `gpt-4o-2024-11-20` as a feasible automated evaluator in our framework. Details are provided in Appendix D.4. For evaluation metrics, following prior work [22, 37], AGENTIF adopts two metrics for evaluation: constraint success rate (**CSR**) and instruction success rate (**ISR**). CSR measures the proportion of individual constraints that are correctly satisfied by the model's response. For a given instruction $i$, $C_i$ is the number of constraints associated with it, and $c_{i,j}$ indicates whether the $j$-th constraint in instruction $i$ is satisfied. ISR measures the proportion of instructions for which all constraints are satisfied. Supposing $N$ is the number of instruction:

$$\text{CSR} = \frac{\sum_{i=1}^{N} \sum_{j=1}^{C_i} \mathbb{1}\left[c_{i,j} = \texttt{satisfied}\right]}{\sum_{i=1}^{N} C_i}; \quad \text{ISR} = \frac{\sum_{i=1}^{N} \mathbb{1}\left[\bigwedge_{j=1}^{C_i}(c_{i,j} = \texttt{satisfied})\right]}{N}$$

## 4 Experiments

In this section, we introduce the experiments and empirical analyses on AGENTIF, including experimental setup (§ 4.1), main results on AGENTIF (§ 4.2), error analysis (§ 4.3), analysis on instruction length (§ 4.4) and meta constraints (§ 4.5). More experimental analyses are presented in Appendix F.

### 4.1 Experimental Setup

We evaluate various advanced LLMs on AGENTIF, including non-thinking models: GPT-4o [11], DeepSeek-V3 [17], Claude 3.5 Sonnet [1], LLaMA 3.1 Series [5], Qwen3 [26], and Mistral [13]; thinking models: o1-mini [12], QwQ 32B [25], GLM-Z1 32B [27], DeepSeek-R1 [6], and DeepSeek-R1 distilled models [6]; and academic models developed specifically for instruction following, including Crab [21] and Conifer [24]. For all models, we set the sampling temperature to $0$. For reasoning models, we remove intermediate reasoning tokens and evaluate only the final response.

### 4.2 Main Results

All experimental results are shown in Table 2. We can observe that:

(1) **All models demonstrate suboptimal performance.** Even the best-performing model, o1-mini, achieves only a CSR of $59.8$. ISR results are even lower, with the highest reaching just $27.2$. Compared to their performance on the commonly used benchmark IFEval [37], all models exhibit a dramatic drop, for example, GPT-4o drops from $87.0$ to $58.5$. This indicates that existing models are still far from perfectly following constraints in agentic scenarios. AGENTIF poses a significant challenge to existing LLMs, highlighting the need for further research in instruction following under real-world agentic scenarios. As instruction following is a prerequisite ability for building reliable

| Models | Presentation Type | | | Type | | | ISR | CSR |
|---|---|---|---|---|---|---|---|---|
| | Vanilla | Condition | Example | Formatting | Semantic | Tool | | |
| [T]GPT-5 | 61.6 | 33.7 | 80.0 | 64.7 | 61.7 | 45.6 | 30.2 | 60.8 |
| [T]GLM-4.6 | 59.2 | 42.2 | 87.2 | 62.1 | 62.1 | 49.9 | 23.1 | 60.5 |
| [N]Claude-4-Sonnet | 59.1 | 44.3 | 83.6 | 62.2 | 62.2 | 45.5 | 21.5 | 60.1 |
| [N]Claude-3-7-Sonnet | 60.9 | 38.9 | 69.2 | 60.1 | 61.3 | 50.9 | 23.3 | 59.5 |
| [N]GPT-OSS-120B | 60.9 | 34.1 | 61.5 | 57.8 | 61.2 | 50.4 | 29.6 | 58.4 |
| [T]o1-mini | 59.8 | 37.5 | 80.8 | 66.1 | 59.1 | 43.2 | 26.9 | 59.8 |
| [N]Claude-3-7-Sonnet | 60.9 | 38.9 | 69.2 | 60.1 | 61.3 | 50.9 | 23.3 | 59.5 |
| [N]GPT-4o | 58.0 | 35.1 | 80.8 | 65.8 | 56.5 | 43.2 | 26.4 | 58.5 |
| [T]Qwen3-32B | 57.5 | 41.1 | 80.6 | 57.7 | 62.5 | 45.7 | 24.9 | 58.4 |
| [T]QwQ-32B | 57.5 | 35.6 | 82.7 | 61.4 | 59.4 | 43.2 | 27.2 | 58.1 |
| [T]DeepSeek-R1 | 56.1 | 41.4 | 87.0 | 61.4 | 58.9 | 44.4 | 22.2 | 57.9 |
| [T]GLM-Z1-32B | 56.7 | 37.9 | 83.6 | 60.2 | 59.6 | 43.1 | 23.8 | 57.8 |
| [N]DeepSeek-V3 | 54.9 | 41.5 | 84.5 | 59.3 | 58.9 | 40.8 | 21.9 | 56.7 |
| [N]Claude-3-5-Sonnet | 57.3 | 36.9 | 69.2 | 61.5 | 56.0 | 43.3 | 24.9 | 56.6 |
| [N]Meta-Llama-3.1-70B-Instruct | 55.1 | 35.0 | 84.3 | 61.6 | 55.6 | 42.8 | 20.9 | 56.3 |
| [T]DeepSeek-R1-Distill-Qwen-32B | 54.5 | 39.6 | 73.1 | 55.7 | 57.2 | 45.2 | 20.7 | 55.1 |
| [T]DeepSeek-R1-Distill-Llama-70B | 55.4 | 37.7 | 69.2 | 56.5 | 56.6 | 44.1 | 19.9 | 55.0 |
| [N]Meta-Llama-3.1-8B-Instruct | 53.5 | 36.6 | 71.4 | 55.6 | 54.8 | 43.5 | 19.9 | 53.6 |
| [S]Crab-DPO-7B | 48.3 | 24.3 | 57.5 | 48.8 | 47.4 | 41.9 | 10.1 | 47.2 |
| [N]Mistral-7B-Instruct-v0.3 | 47.9 | 29.2 | 53.8 | 47.0 | 48.6 | 39.8 | 11.5 | 46.8 |
| [S]Conifer-DPO-7B | 45.6 | 27.0 | 50.5 | 42.0 | 46.9 | 41.8 | 10.7 | 44.3 |

Table 2: Success rates (%) of various proprietary and open-source LLMs on AGENTIF, sorted by CSR in descending order. [N] denotes non-thinking models, [T] denotes thinking models, and [S] denotes models explicitly designed for instruction following by the academic community. As described in § 3.4, CSR indicates the proportion of correctly followed individual constraints, and ISR presents the proportion of instructions in which all constraints are satisfied.

LLM-based agents, we argue that prior to developing agents, it is crucial to evaluate the fundamental instruction-following abilities of LLMs to inform effective prompt and workflow design.

(2) For specific constraint types, models perform much lower on the condition and tool constraints. Compared to vanilla constraints, condition constraints require an additional step to determine whether the constraint should be triggered. Thus, the low success rate on condition constraints may be that the model struggle in correctly determining whether the condition is triggered. As condition-based constraints account for approximately $42.6\%$ of real-world applications and are usually overlooked in existing instruction-following research [21, 24], we advocate for increased efforts to handle conditional instructions. For tool constraints, failures may stem from issues such as missing required parameters or failing to invoke the specified tools. Detailed error analysis is provided in § 4.3. The primary reason may be that the models fails in handling specification-heavy tasks [20, 2] and tool usage typically involves complex specifications. As tool usage is necessary in agentic applications, we call for more attention to tool specification adherence to enable more reliable agent behavior. For example constraints, models perform relatively better. It suggests that they can effectively infer and meet requirements when provided with in-context examples, indicating that providing concrete examples in prompt design can facilitate better understanding and imitation of desired behaviors.

(3) Across different models, we find that thinking models generally perform better, suggesting that complex instruction following also requires reasoning capabilities and that test-time scaling also benefits. In contrast, models specifically trained for instruction following by the academic community perform worse. This may be due to their primary focus on constructing SFT datasets for training base models [21, 24]. Given that industry models, such as Llama 3, already demonstrate strong performance due to using large-scale SFT data, we encourage the academic community to explore more advanced approaches, such as reinforcement learning [15], which has recently proven effective for enhancing model capabilities [6] but remains under-explored in instruction following.

In conclusion, AGENTIF poses significant challenges for existing models, particularly on constraints commonly used in agentic applications, such as tool constraints. We call for increased research efforts to improve instruction following in agentic scenarios.

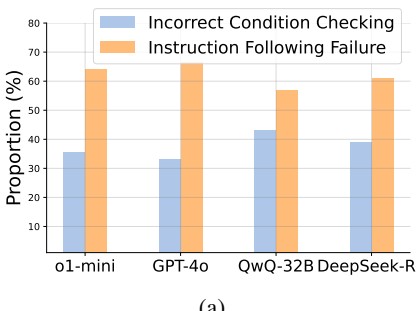 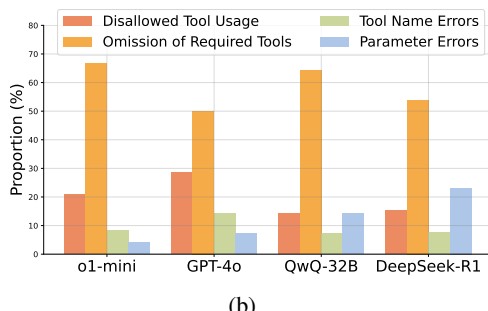

(a)        (b)

Figure 4: Error proportions (%) on condition and tool constraints. **Figure (a)** shows the errors in handling condition constraints, including condition check failure, where the model fails to recognize the condition, and constraint following failure. **Figure (b)** shows the errors from tool constraints, including disallowed tool usage (utilizing explicitly prohibited tools), omission of required tools (failing to employ required tools), tool name errors (invoking non-existent or incorrect tools), and parameter errors (applying incorrect or illegal arguments).

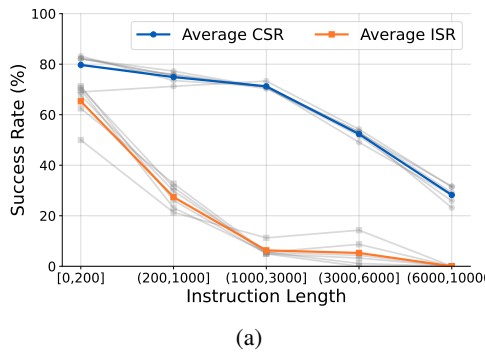 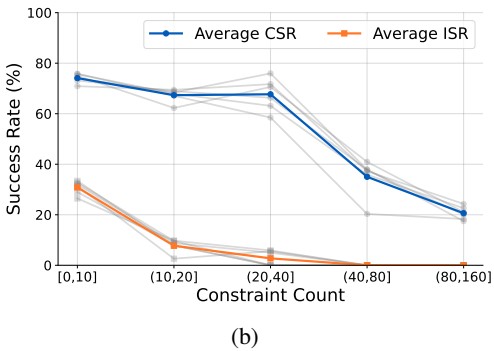

(a)        (b)

Figure 5: Success rates on instructions with varying length or constraint counts. Gray lines show results of the top 6 models in Figure 2, and the colored lines present their average.

## 4.3 Error Analysis

As shown in Figure 2, LLMs perform particularly poorly on two types of common constraints in agentic scenarios: condition and tool constraints. Therefore, we conduct a detailed error analysis on these cases. Specifically, we analyze the errors of the four representative LLMs, including o1-mini, GPT-4o, QwQ-32B, and DeepSeek-R1, and manually investigate their error types.

**Analysis on Condition Constraint**     We identify two main types of failure in following condition constraints: (1) incorrect condition checking, where the LLM fails to determine whether a condition is triggered; and (2) instruction following failure, where the LLM fails to follow the constraint even when the condition is triggered. We conduct a controlled experiment to assess the relative proportion of the two causes. Specifically, we select all the failed conditional constraints from each investigated model, remove their conditional components, and then convert them into vanilla constraints that must be met while keeping all other elements unchanged. If the model then succeeds, it indicates an error in condition checking; if it still fails, it suggests a general failure to follow the constraint. The results are shown in Figure 4(a). We can observe that a substantial portion (above $30\%$) of errors are due to incorrect condition checks, suggesting that condition constraints introduce new challenges in determining whether a constraint is triggered. Since existing work on instruction following usually overlooks such conditional constraints, we advocate constructing targeted post-training data.

**Analysis on Tool Constraint**     We conduct error analysis on tool constraint violations. Specifically, we sample 50 tool usage errors from each investigated model and identify four primary error cate-

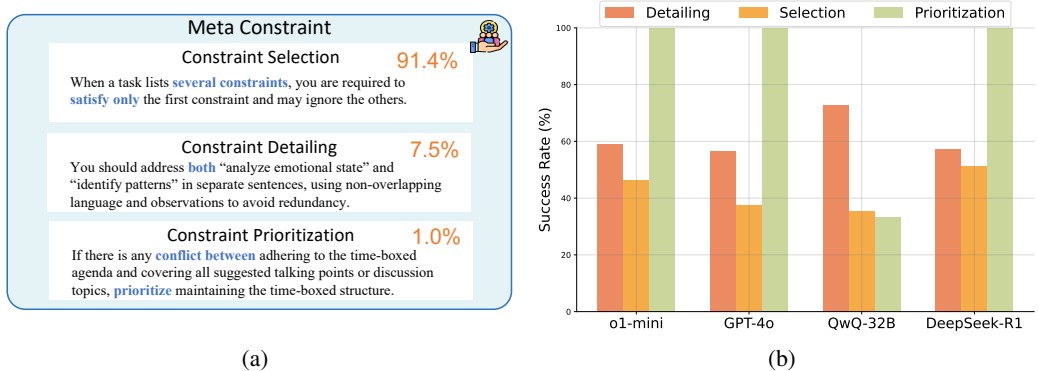

(a)                                                                                    (b)

Figure 6: **Figure (a)** illustrates three types of meta constraints and examples. Most meta constraints fall within the Constraint Selection category, which requires models to follow one specific constraint. **Figure (b)** presents the success rates of different investigated models on each meta constraint type.

gories: disallowed tool usage, omission of required tools, tool name errors, and parameter errors. The error proportions are shown in Figure 4(b). We observe that disallowed tool usage and omission of required tools constitute the primary errors, suggesting that tool invocation decision-making remains challenging for models. A portion of errors also stems from non-compliance with tool usage specifications, exhibited as incorrect tool names or parameters. Notably, we observe an interesting phenomenon: thinking models more frequently neglect required tools. The reason may be that the thinking models may tend to rely more on their internal knowledge. We encourage the research community to conduct further investigations into these specific underlying causative mechanisms.

## 4.4    Analysis of Instruction Length and Constraint Quantity

While prior work has shown that long texts pose significant challenges for LLMs, most studies have focused on long textual contexts within the query [2], such as long document question answering. Little work investigates the challenges arising from long instructions, which may be due to the lack of evaluation data with lengthy instructions. AGENTIF provides such an evaluation platform, with instructions averaging $1,723$ words and containing about $11.9$ constraints each. We analyze model performance on instructions of varying lengths and constraint counts in AGENTIF. We bucket the data by instruction length or number of constraints, then compute the success rates within each bucket. The results are shown in Figure 5. We observe that model performance generally declines with increasing instruction length or constraint count, indicating that longer instructions or those with more constraints are inherently more difficult, which is consistent with the intuition. Notably, when instruction length exceeds $6,000$ words, the ISR scores of all models are nearly $0$. This indicates that overly long instructions are rarely followed perfectly and should be avoided in practice. Instead, one can explore decomposing tasks into several sub-tasks with several shorter instructions to alleviate instruction following failures [35]. We call for more research efforts to enhance models' ability to follow long instructions. As discussed in [20], the primary reason LLMs fail on specification-heavy instructions is the in-context learning limitation. A promising direction is collecting post-training data with long instructions, which remains underexplored due to the scarcity of such data. One potential source is manuals[6], such as camera manuals, which can serve as long instructions and be used to automatically construct question-answer pairs for post-training. We leave this for future work.

## 4.5    Analysis of Meta Constraints

We observe a prevalent type of constraint in AGENTIF, which we refer to as **meta constraints**. Unlike regular constraints that apply directly to the model's response, meta constraints govern other constraints. We find approximately $25\%$ of instructions in AGENTIF include meta constraints. As shown in Figure 6(a), we categorize them into three types: (1) constraint selection, where the meta constraint requires the model to follow only a specific constraint; (2) constraint detailing, where it adds

---
[6]https://manymanuals.com/

further requirements to an existing constraint; and (3) constraint prioritization, where it defines the relative priority among multiple constraints. Figure 6(b) illustrates the success rates of different meta constraints. We can observe that the models generally perform the worst on constraint selection. One possible reason is that the meta constraint may conflict with the original constraints, which confuses LLMs. Future work may explore giving meta constraints higher priority to improve compliance [28], while carefully mitigating potential safety risks such as prompt injection attacks [18].

# 5 Conclusion

In this paper, we present AGENTIF, the first instruction-following benchmark for agentic scenarios. AGENTIF comprises 707 instructions across 50 real-world agentic applications. Each instruction has an average length of $1,717$ tokens and includes approximately $11.9$ constraints, covering a diverse range such as condition and tool constraints. We evaluate various representative and advanced LLMs on AGENTIF and find that current models generally perform poorly, and the best model perfectly follows fewer than $30\%$ of the instructions, which suggests that AGENTIF poses a significant challenge. We further conduct analytical experiments to investigate the failure modes. We find that condition and tool constraints introduce new challenges. We also observe performance degradation as instruction length increases, which aligns with the intuition that longer instructions are more difficult. We encourage more research efforts to enhance instruction-following capabilities in agentic scenarios.

# Acknowledgement

This work is supported by the National Natural Science Foundation of China (No. 52539001), the Beijing Natural Science Foundation (No. L243006), and partially by CHN Energy Dadu River Big Data Services Co., Ltd.

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

# Appendices

## A   Limitations

We discuss the limitations as follows: (1) Although the construction of AGENTIF is semi-automated, it still requires substantial manual verification, which limits its direct generalization to generating a large scale of data. In the future, we plan to explore automated methods for constructing post-training data to enhance instruction-following capabilities. (2) AGENTIF includes instructions only in Chinese and English, lacking broader multilingual coverage, which may limit its broader usage. We encourage the community to extend the dataset to support more languages. (3) All experiments are conducted in a zero-shot setting, and we do not explore prompt engineering techniques such as in-context demonstrations. As this work focuses on dataset construction and model evaluation, we leave prompt engineering for future work.

## B   Ethical Considerations

We discuss ethical concerns and broader impacts as follows: (1) **Intellectual Property.** The agents obtained from GitHub repositories are shared under GPL-3.0[7] or Apache-2.0 licenses[8]. We strictly adhere to the respective licensing terms, and all data are used solely for academic research. For industrial agents, we obtained approval from internal review boards to use and release the data under the GPL-3.0 license. These industrial applications target enterprise users in China, and we only collect statistics on active users and service requests and do not conduct user behavior analysis. Notably, the agents include only system prompts without any user queries to prevent information leakage. Instead, all user queries in AGENTIF are created by hired annotators. AGENTIF will be released under the GPL-3.0 license. (2) **Broader Impacts.** This work aims to construct a benchmark for evaluating instruction-following in agentic scenarios. Leaderboard rankings on AGENTIF should not be used for adversarial comparisons or interpreted as evidence of misconduct in other research efforts. The benchmark data should not be incorporated into the training process of LLMs to avoid potential contamination or leakage. (3) **Controlling Potential Risks**. All data have been subjected to rigorous safety checks and are well anonymized to eliminate sensitive content. (4) **Human Annotation.** We employ eight annotators (gender-balanced) for data annotation and verification. All annotators are fairly paid based on workload and are informed of the intended use of the data before annotation. No personal information of annotators is involved in the dataset. (5) **LLM Usage.** We used GPT-4o and Claude 3.7 Sonnet for paraphrasing some sentences of this paper.

## C   Details of Constraint Taxonomy

### C.1   Constraint Type

This section provides a detailed overview of our constraint taxonomy. Figure 7 illustrates the distribution of constraints across different categories.

**Formatting Constraints**   Formatting Constraints dictate the structural form and presentation of model outputs. They ensure responses conform to a specific format, layout, or visual style, vital for machine readability or subsequent processing. Examples include: (1) Syntax Formatting: requiring outputs in formats like JSON, XML, or Markdown. (2) Layout Structure: specifying bullet points, tables, or paragraph length. (3) Symbol Conventions: enforcing specific symbols like date patterns or currency symbols. (4) Instructional Structure: mandating specific response organizations, such as "explain the concept in three steps."

**Semantic Constraints**   Semantic Constraints govern the factual correctness, informativeness, and intended meaning of model outputs. They ensure content aligns with task requirements and includes essential semantic elements. Common types are: (1) Content Targeting: restricting the output's topic or focus. (2) Information Completeness: requiring specific elements, such as time, location, or

---

[7]https://www.gnu.org/licenses/gpl-3.0.en.html
[8]https://www.apache.org/licenses/LICENSE-2.0

people. (3) Keyword Presence: mandating the inclusion of particular phrases or terms. (4) Stylistic Pequirements: dictating aspects like "use language understandable to children."

**Tool Constraints**   Tool Constraints restrict the computational or external resources a model can use when generating responses. These are crucial for simulating specific environments or adhering to usage restrictions. Examples include: (1) Tool Usage: restricting callable functions or external tools, e.g., "only built-in Python functions may be used." (2) Computational Limitations: imposing resource constraints, such asdo not use GPU acceleration."

## C.2   Constraint Presentation Type

The constraint presentation type refers to how constraints are conveyed to the model within the instruction. We categorize this type into three distinct forms:

**Vanilla Constraints**   are stated directly and unconditionally in the prompt. These constraints are always in effect, regardless of the input content or model behavior. For example, "Answer in Chinese", i.e., the model must always respond in Chinese, regardless of the question.

**Condition Constraints**   are activated only when specific conditions are met, either from the user input or the model's output. For instance, a constraint like "flag the response if it contains sensitive content" depends on input triggers, while "apply markdown formatting only when markdown is used" relies on the model's behavioral context.

**Example Constraints**   are not explicitly stated but implied through in-context demonstrations. For example, providing an example written in Shakespearean English implicitly requires the model to generate responses in a similar style, or showing few-shot outputs in structured JSON format signals that the same structure should be followed.

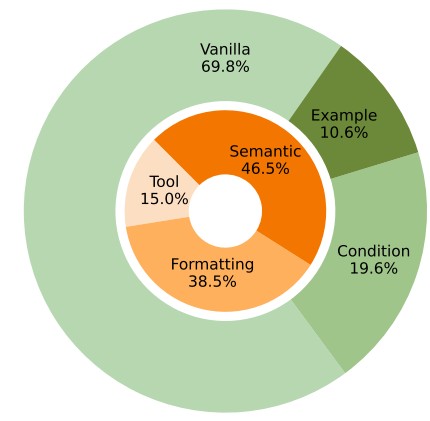

Figure 7: Distribution of constraint types. The inner ring shows the breakdown of constraint types into Semantic, Formatting, and Tool categories. The outer ring further categorizes constraints based on their presentation type, including Vanilla, Condition, and Example. Semantic constraints and vanilla activation are the most prevalent in the dataset.

# D   Detailed Information about AGENTIF

## D.1   Prompts for Automatic Annotation During Dataset Construction

Table 6 provides the prompt template for instruction collection. Templates for constraint annotation are found in Tables 7 and 8. In the evaluation generation phrase, the template for generating conditional checks is in Table 9. Finally, Tables 11, 13, and 12 detail the prompt templates for generating three types of evaluation.

## D.2   Detailed Information about Human Annotation During Dataset Construction

We invited graduate students from the Department of Computer Science to participate in the data annotation process, and fairly compensated them based on pre-agreed salaries and workloads. All employment was formalized through contracts and conducted in full compliance with local regulations. During the annotation process, we conducted three rounds of sampled review and feedback to iteratively refine and finalize the high-quality annotation results.

| Category | Application Domain | % of Dataset |
|---|---|---|
| General | General Question Answering | 15.3% |
| | Knowledge Reasoning | 8.5% |
| | Information Retrieval and Search Optimization | 1.6% |
| | Conversational AI Assistant | 1.6% |
| | General Task Agent | 1.4% |
| Health | Mental Health Support | 10.9% |
| | Health Advisory | 1.6% |
| Legal | Law Question Answering | 9.9% |
| Tourism | Audio Tour | 8.2% |
| Coding | Code Collaboration | 7.2% |
| | Software Engineering | 1.6% |
| | Web App Development | 1.4% |
| | CLI-based Coding Assistant | 1.4% |
| Gaming | Game Design | 6.2% |
| Research | Research Assistant | 4.7% |
| | News Intelligence | 3.1% |
| | Research Writing | 1.6% |
| Enterprise | Enterprise Strategy And Meeting Intelligence | 3.5% |
| | Enterprise Operations | 3.1% |
| | Lead Generation | 1.6% |
| | Market Strategy | 0.4% |
| Gaming | Game AI | 3.1% |
| Finance | Personal Finance Management | 2.3% |

Table 3: Distribution of categories and application domains in AGENTIF.

### D.3 Detailed Information about Dataset Diversity

Our benchmark is constructed from agentic instructions collected across 50 representative real-world agent applications, sourced from both industrial use cases and GitHub repositories. These tasks were carefully selected to evaluate LLMs' instruction-following capabilities in a broad range of scenarios.

Specifically, the dataset spans 9 categories (General, Legal, Health, Coding, Finance, Research, Enterprise, Tourism, Gaming), 23 application domains, as shown in the Table 3. We also welcome community contributions of additional agent system prompts to expand our evaluation.

### D.4 Agreement Between the Automatic Evaluation Method and Human Evaluation

LLM-based evaluation is widely used and has proven effective across domains. Our protocol combines code-based checks with GPT-4o judgments, and we validate the latter via human review. Concretely, we randomly sample 200 responses from the top five models (o1-mini, GPT-4o, Qwen3-32B, QwQ-32B, DeepSeek-R1) and manually assess whether each response satisfies the stated constraints. The agreement between GPT-4o and human judgment is 94% (200 samples), indicating that GPT-4o provides sufficiently reliable automated evaluation for our setting.

## E  Experimental Details

**Evaluation Models.**  We conduct our evaluation using a diverse set of language models, including o1-mini, GPT-4o (2024-11-20), Qwen3-32B, QwQ-32B, DeepSeek-R1, GLM-Z1-32B (0414), DeepSeek-v3 (2024-03-25), Claude 3.5 Sonnet (2024-10-22), Meta-LLaMA-3.1-70B-Instruct, DeepSeek-R1-Distill-Qwen-32B, DeepSeek-R1-Distill-LLaMA-70B, Meta-LLaMA-3.1-8B-Instruct, Mistral-Crab-DPO-7B, Mistral-7B-Instruct-v0.3, and Mistral-Conifer-DPO-7B.

| Model | still (%) | correct → wrong (%) | wrong → correct (%) |
|---|---|---|---|
| GPT-4o | 91.2 | 3.9 | 2.1 |
| QwQ-32B | 88.9 | 4.6 | 5.9 |
| Claude-3-5-Sonnet | 72.5 | 8.4 | 16.9 |

Table 4: Effect of adding explicit constraints to example-based instructions.

| Model | Multi-Turn CSR (%) | Single-Turn CSR (%) |
|---|---|---|
| o1-mini | 64.6 | 77.3 |
| GPT-4o | 66.1 | 72.3 |
| Qwen3-32B | 59.6 | 77.1 |
| QwQ-32B | 57.6 | 76.7 |
| DeepSeek-R1 | 63.1 | 81.0 |

Table 5: Comparison of CSR on multi-turn versus single-turn instructions.

**Experimental Hyperparameters.** In all experiments, we set the temperature to $0$ to ensure reproducibility. The maximum number of generated tokens is set to $32,000$. For models with a context length shorter than $32,000$ tokens, we set their maximum allowable context length to match the `max-token` limit.

**Experiment Cost.** We use `gpt-4o-2024-11-20` as the evaluator throughout our experiments. Each evaluation round costs approximately $20.

# F  Further Experimental Analysis

## F.1  Analysis of Example Constraint Type

We analyze errors related to instructions that include in-context learning examples (i.e., example constraints). Specifically, **we manually add the corresponding explicit constraint for each example constraint**. We observe three patterns: *still*, where performance remains unchanged; *correct →* *wrong*, where the model fails to follow the constraint after adding the explicit constraint; and *wrong → correct*, where the model follows the constraint after the explicit constraint is added.

As shown in Table 4, adding explicit constraints improves the model's instruction-following ability. This suggests that the primary challenge for current models lies in inferring implicit constraints from examples rather than executing explicit ones.

## F.2  Analysis of Multi-Turn Data in AGENTIF

Our benchmark includes multi-turn instructions that depend on dialogue history, accounting for $19.1\%$ ($135/700$) of the instructions.

We compare performance on multi-turn versus single-turn instructions. As shown in Table 5, models follow multi-turn instructions markedly worse than single-turn ones, which is intuitive given the longer-context understanding required in multi-turn settings.

You are given two pieces of information:
1. A **Task Description** summarizing what the agent is designed to do.
2. **Input Variable Annotations** listing the input variables, each with a brief explanation.

**Your task is to:**

- Generate multiple sets of variable content that are rich, detailed, and expanded. Each set must include specific, meaningful, and realistic information, maximizing substance while staying coherent.

- Ensure strong diversity across examples, including both speaking styles (storytelling, poetic, formal, humorous, dramatic, etc.) and imagined scenarios (different realistic scenes that meaningfully shape the content).

- Match the meaning of each variable based on its comment from Input Variable Annotations.

- Use the Task Description to guide the overall theme and content of the generated examples.

**Input:**

{input}

**Output Format:**
Return a **JSON array** where:

- Each item is a JSON object (a dict) corresponding to one complete example.

- The keys of each object exactly match the variable names listed in the Input Variable Annotations.

Example structure:

```
[
  {
    "variable_name_1": "<filled content>",
    "variable_name_2": "<filled content>"
  },
  {
    "variable_name_1": "<filled content>",
    "variable_name_2": "<filled content>"
  }
]
```

Table 6: Prompt for query generation in instruction collection.

You are given a **system prompt**. Your job is to extract atomic constraints that apply specifically to the **expected response** generated by the model, as dictated by the system prompt.

**Please follow the instructions below precisely:**

- Only extract constraints that apply to the **response**, not those describing or constraining the input variables or instructions to the user.

- Read the system prompt line by line and extract the **smallest possible atomic constraint units** from any content that imposes rules, structure, or expectations on the response.

- This includes:
    - Explicit instructions such as: "Your task is to...", "You must...", "Please do the following..."
    - Numbered lists of required actions
    - Formatting, style, or output structure expectations
    - Content rules or restrictions
    - Demonstrations or examples that implicitly define how the model should respond (e.g., few-shot examples, response templates, or stylistic samples). In these cases, extract the implied constraint as faithfully as possible (e.g., "Respond in JSON format", "Follow the narrative style shown").

- For each constraint, determine whether it is:
    - "vanilla": The rule applies to all responses regardless of input or task branch.
    - "conditional": The rule only applies in certain contexts (e.g., a certain kind of task or response type).
    - "example": The rule is not explicitly stated, but implied from a given example or demonstration in the prompt.

- Sometimes the system prompt gives a rule or behavior without explicitly stating the condition, but it's only meant to apply in a certain type of response. In such cases, you should **infer the missing condition** and rewrite the constraint in the form If [condition], then [rule]. These should be marked as "conditional" even if no "if" appears in the prompt.

- When the prompt contains **an example**, you should recognize that it implies certain response expectations, and you are expected to infer the corresponding constraints accordingly.

- Do **not rewrite or generalize** the constraint. Extract the **exact wording** from the prompt wherever possible.

- Return your answer as a list of dictionaries, where each dictionary contains:
    - desc: the extracted constraint (verbatim from the prompt, or inferred if conditional)
    - dimension: one of "unconditional", "conditional", or "example"

- If the system prompt contains no constraints on the response, return an empty list: []

**Example Output:**

```
[
  {
    "desc": "Always use \"you\" and \"your\" when addressing the user.",
    "dimension": "vanilla"
  },
  {
    "desc": "If no symptoms are reported, explain why further screening is still
        necessary.",
    "dimension": "conditional"
  }
]
```

**Input System Prompt:**

{prompt}

Table 7: Prompt for extracting atom-level constraints from block instructions in constraint annotation.

You are given a single constraint. Your task is to classify this constraint into one or more of the following categories. Prefer to choose **only one** category unless you believe the constraint **clearly fits multiple types**. After classification, provide a **brief explanation** supporting your decision.

**Constraint Categories:**

1. **formatting** — *Controls the structure or presentation format of the output.*
   Examples include:

   - Syntax format (e.g., JSON, XML, Markdown)
   - Layout and structure (e.g., bullet points, tables, paragraph length)
   - Symbol and notation norms (e.g., date format "YYYY-MM-DD", currency symbol "¥")
   - Interaction steps (e.g., "explain the principle in three steps")

   *Example:* "Present the result using LaTeX math notation"

2. **semantic** — *Ensures the output content is meaningful, accurate, and complete.*
   Examples include:

   - Factual accuracy (e.g., no fabricated data)
   - Logical consistency
   - Information completeness (must include specified elements)
   - Keyword requirements (must contain specified terms)
   - Style or tone (e.g., "explain in child-friendly language")
   - Neutrality of position (e.g., "avoid emotionally charged language")
   - Terminology standards (e.g., "use ISO names for chemicals")

   *Example:* "The answer must include the event's time, location, and key figures"

3. **Tool** — *Limits the usage of computational resources or external dependencies.*
   Examples include:

   - Data source limitations (e.g., "only use data from after 2020")
   - Computational restrictions (e.g., "no GPU acceleration")
   - Tool/library restrictions (e.g., "use only built-in Python functions")

   *Example:* "Do not access any online resources during analysis"

**Output Format:**

```
{
  "type_list": ["Your chosen category or categories"],
  "explanation": "Your reasoning for choosing this classification"
}
```

**Input Constraint:**

```
{constraint}
```

Table 8: Prompt for classifying constraints into formatting, semantic, or tool categories in constraint annotation.

You are a helpful assistant that specializes in constraint verification.
Your task is to process a **conditional constraint** and produce two outputs:

1. A **yes/no question** that can be used to verify whether the **condition** is satisfied.

2. The **main constraint** that should be enforced if the condition is true. This should exclude the conditional part and be expressed as a standalone, unconditional constraint.

**Please follow these instructions:**

- If the condition refers to the **input query**, the question should focus on analyzing the input.

- If the condition refers to the **response**, the question should focus on analyzing the response.

- Keep the extracted main constraint faithful to the original meaning, but **remove the conditional clause** (e.g., remove "If..." or "When...").

**Return your output as a JSON dictionary with the following keys:**

```
{
  "condition_check_question": "{Your yes/no question}? Please answer YES/NO directly and
      do not enter anything else.",
  "main_constraint": "{The unconditional constraint to verify if the condition is true
      .}"
}
```

**Note:** The constraint itself is the primary basis for generation. The instruction paragraph is provided only as auxiliary context, and should be used only when the constraint alone is ambiguous or underspecified.
**Input**
Here is the full instruction paragraph where the constraint appears: {instruction}
The constraint to verify: {constraint}

Table 9: Prompt for decomposing a conditional constraint into a condition-checking question and a standalone constraint in evaluation generation.

You are an expert at analyzing natural language constraints and determining how they can be verified.
Your task is to **classify a given constraint** based on whether it can be validated:

1. code — Directly by code

2. llm_assisted_code — By code after extracting needed content via LLM

3. llm — Only by using LLM to semantically assess it

**Please follow these guidelines:**

- If the constraint can be validated by simple logic (e.g., length, presence, format) and the content is directly accessible from the response, classify it as code.

- If the constraint requires extracting a specific section from the response (e.g., "intro", "conclusion", "step 1") before performing validation (e.g., counting words), classify it as llm_assisted_code.

- If the constraint requires open-ended, semantic, or subjective understanding (e.g., logical correctness, relevance, tone, or fact-checking), classify it as llm.

**Return your answer *only* as a JSON dictionary in the following format:**

```
{
  "constraint_type": "code" | "llm_assisted_code" | "llm",
  "explanation": "Your brief reasoning here"
}
```

**Note:** The constraint itself is the primary basis for classification. The instruction paragraph is provided only as auxiliary context, and should be used only when the constraint alone is ambiguous or underspecified.
**Input**
Here is the full instruction paragraph where the constraint appears: {instruction}
The constraint to classify: {constraint}

Table 10: Prompt for classifying constraints by their verifiability type: directly by code, LLM-assisted code, or purely by LLM.

You are a helpful assistant that specializes in verifying whether model responses comply with specific constraints. Your task is to **generate a yes/no question** that can be used to determine whether a model response satisfies a given constraint. This question should be phrased so that an LLM (or a human evaluator) could answer it just by reading the model's response.

**Please follow these rules:**

1. The question should be **clear, specific, and binary** — it should be answerable with "yes" or "no".

2. It must refer explicitly to what the constraint is checking (e.g., structure, length, tone, factuality).

3. If the constraint refers to a specific section (e.g., "intro", "step 1", "conclusion"), include that in the question.

**Return your answer in the following format:**

```
{
  "validation_question": "{Your full yes/no question here}? Please answer YES/NO
      directly and do not enter anything else."
}
```

**Note:** The constraint itself is the primary basis for generation. The instruction paragraph is provided only as auxiliary context, and should be used only when the constraint alone is ambiguous or underspecified.
**Instruction Paragraph (Context):**
This is the full instruction paragraph that provides context for the constraint: {instruction}
**Constraint to Verify:** {constraint}

Table 11: Prompt for generating a yes/no validation question from a constraint.

You are a helpful assistant that specializes in generating extraction instructions to support constraint verification. Your task is to generate a concise and precise instruction that tells an LLM **what specific part of the response needs to be extracted**, so that the extracted content can later be verified by code.
**Please follow these guidelines:**

1. The instruction should clearly specify **what** to extract from the response (e.g., "Extract the introduction part", "Extract the function used", "Extract the final answer sentence").

2. Base your output on the constraint provided below.

3. You may refer to the instruction paragraph **only when the constraint is ambiguous** and requires context.

4. Return your output as a JSON dictionary in the following format:

```
{
  "extraction_instruction": "{your generated extraction instruction}. Return the
      extracted content verbatim from the response. If multiple segments are found,
      return them as a Python-style list of strings. If nothing is found, return None."
}
```

**Input:**
**Instruction Paragraph:**
Here is the full instruction paragraph where the constraint appears: {instruction}
**Constraint to support:** {constraint}
**Helper analysis:** {helper_analysis}

Table 12: Prompt for generating extraction instructions from a constraint to support code-based verification in hybrid evaluation.

You are tasked with implementing a Python function named `check_following` that determines **whether a given response satisfies a specified constraint**. The function must return `True` if the constraint is satisfied and `False` otherwise.

**Requirements:**

- The function accepts only one parameter: `response`, which is a Python string.

- The function must return a boolean value (`True` or `False`) based on whether the `response` adheres to the constraint.

- The function must not include any I/O operations, such as `input()` or `ArgumentParser`.

- The Python code for constraint verification should be designed to be generalizable, e.g., using regular expressions or other suitable techniques.

- Only return the exact Python code, with no additional explanations.

**Note:** The constraint itself is the **primary basis** for classification. The instruction paragraph is provided **only as auxiliary context**, and should be used only when the constraint alone is ambiguous or underspecified.

**Instruction Paragraph:**
Here is the full instruction paragraph where the constraint appears: {instruction}
**Specific Constraint to Verify:** {constraint}

Table 13: Prompt for generating a Python function that verifies whether a response satisfies a specified constraint.

