# OpenReview forum: "AGENTIF: Benchmarking Large Language Models Instruction Following Ability in Agentic Scenarios"
_NeurIPS.cc/2025/Datasets_and_Benchmarks_Track — NeurIPS 2025 Datasets and Benchmarks Track spotlight_

### Official Review · Reviewer_uQoP · 2025-06-23

**Rating:** 5
**Confidence:** 2

**Summary:**

This paper introduces AGENTIF, a novel benchmark for evaluating the instruction-following capabilities of LLM-based agentic systems. Its key contribution is addressing the shortfalls of existing benchmarks, which typically rely on brief, synthetic instructions that fail to capture the complexity of real-world agent prompts. The authors evaluate several open and closed-source LLMs on this benchmark and find that even advanced models struggle with complex instructions and constraints, highlighting a gap in current agentic AI capabilities.

**Dataset Code Accessibility:**

Yes

**Dataset Code Comments:**

The code and data for the proposed benchmark are available on GitHub and Hugging Face.

**Ethical Comments:**

Generally, the authors have responsibly addressed the ethical aspects of their work and been transparent about its potential societal impacts.

**Ethical Considerations:**

No, there are no or only very minor ethics concerns

**Final Justification:**

I appreciate the authors’ thoughtful responses to my questions. All of my concerns have been addressed. Therefore, I am pleased to raise my score.

**Limitations Weaknesses:**

1. The dataset contains around 700 examples, which is not large. As a result, the statistical confidence of the results might be limited, making the datasets larger could help to make the benchmark more reliable and reduce randomness.

2. This benchmark only evaluates single-turn interactions. In contrast, most real-world agent tasks involve multiple conversational turns and planning over time. Expanding the benchmark to cover these extended dialogues would better reflect real-world agent performance and reveal potential weaknesses in longer-term interactions.

3. The current evaluation framework depends on LLM APIs. These LLMs carry inherent biases shaped by their original training data, which could unintentionally favour or disadvantage the outputs of some specific models. Adding some human-based evaluation results in the experimental section would help to ensure the fairness and reliability of the benchmark.

**Strengths Contributions:**

1. This paper focuses on a critical challenge in instruction-following for agentic LLMs, namely, the scarcity of high-quality, real-world instructions that are lengthy and embedded with complex constraints. Therefore, this paper targets an important and underexplored problem that has clear implications for practical deployment, as existing Agentic LLMs still struggle with long, heterogeneous instructions.

2. The AGENTIF dataset is thoughtfully designed and rigorously challenging, featuring diverse, complex instructions with multiple constraints. Its human-in-the-loop generation process further ensures high quality and relevance.

3. The paper is clearly organised and highly accessible, making it easy for the audience to follow the argument and get the key contributions of this work.

---

> ### Author Rebuttal · Authors · 2025-07-31
>
> Thanks for your valuable comments!
>
> - **Regarding the Concern About Dataset Size（Weakness #1）**
>
>     Thank you for the feedback. The 700 instances contain **8,449 constraints**, and we report both instruction-level and constraint-level accuracy, which we believe provide a stable evaluation of model performance. Additionally, various well-known benchmarks also prioritize data quality. For example, GPQA includes only 448 instances, and IFEval around 500. We apply strict quality control and manual verification, and hence believe our benchmark offers an effective evaluation.
>
> - **Regarding Multi-Turn data（Weakness #2）**
>
>     Apologies for the confusion. Our benchmark **includes multi-turn instructions** that depend on dialogue history, and they account for **19.1% (135/700)** of the instructions. We investigate the performance in multi-turn vs single-turn instructions. The results are shown below. We find that the results of following multi-turn instructions are much lower than those of single-turn instructions. This is intuitive due to the longer context understanding requirement of multi-turn instructions. Thank you for the feedback. We will add these clarifications and results to the paper.
>
>     |  | Multi-Turn CSR | Single-Turn CSR |
>     | --- | --- | --- |
>     | o1-mini | 64.6 | 77.3 |
>     | GPT4o | 66.1 | 72.3 |
>     | Qwen3-32B | 59.6 | 77.1 |
>     | QwQ-32B | 57.6 | 76.7 |
>     | DeepSeek-R1 | 63.1 | 81.0 |
>
> - **Regarding Potential Evaluation Bias（weakness #3）**
>
>     LLM-based evaluation has been widely used and demonstrated to be an effective evaluation approach in various domains [1].
>     Our evaluation combines both code-based evaluation and GPT-4o evaluation. We also manually verify the reliability of GPT-4o's evaluations. Specifically, we first randomly sample 200 responses from the top 4 LLMs, including o1-mini, GPT-4o, Qwen3-32B, and QwQ-32B. We manually assess whether the responses satisfy each constraint in the instructions. **The agreement between GPT-4o and human judgment is 94%**. Therefore, we believe GPT-4o evaluation is sufficiently effective as an automated evaluation.
>
> Thank you for the feedback. We will add these explanations to the paper.
>
> [1] Gu, Jiawei, et al. "A survey on llm-as-a-judge." *arXiv preprint arXiv:2411.15594* (2024).

---

> > ### Comment · Reviewer_uQoP · 2025-08-05
> >
> > I appreciate the authors’ thoughtful responses to my questions. All of my concerns have been addressed. Therefore, I am pleased to raise my score.

---

> > > ### Author Response · Authors · 2025-08-05
> > >
> > > Thank you very much for your positive feedback and for raising your score!

---

### Official Review · Reviewer_P5jt · 2025-07-01

**Rating:** 5
**Confidence:** 4

**Summary:**

This paper proposes AGENTIF, the first benchmark specifically designed to evaluate large language models’ ability to follow instructions in realistic agentic scenarios. It introduces a detailed taxonomy covering multiple constraint types (formatting, semantic, tool-related) and presentation forms (vanilla, conditional, example). The authors systematically evaluate a wide range of advanced LLMs on this benchmark, revealing substantial performance gaps, especially in handling long, complex instructions and strict tool constraints.

**Dataset Code Accessibility:**

Yes

**Dataset Code Comments:**

The authors provide the benchmark code through GitHub and make the dataset available on Hugging Face. Both resources are open-sourced, well-documented, and readily accessible in a usable and reproducible format.

**Ethical Considerations:**

No, there are no or only very minor ethics concerns

**Final Justification:**

Thanks for the rebuttal and additional experimental results, which have tackled most of my concerns.

**Limitations Weaknesses:**

- **Limited Scope of Tool Constraints**
The current tool constraints primarily address basic requirements (e.g., parameter type validation and invocation syntax) but fail to capture more sophisticated real-world scenarios such as multi-tool orchestration or runtime-adaptive tool selection.  This narrow focus significantly restricts the benchmark's ability to evaluate complex agent workflows.  Future iterations could be enhanced by incorporating dynamic environment simulations (e.g., API rate limiting, network failure handling) and expanding domain coverage to include specialized professional applications.

- **Potential Evaluation Circularity**
While the paper presents a thoughtful and comprehensive evaluation framework, a potential limitation is the reliance on GPT-4o both to determine evaluation methods and to serve as the primary evaluator in LLM-based and hybrid assessments.  Since GPT-4o is also one of the evaluated models, this setup may introduce potential bias or circularity.  Explicitly acknowledging this limitation and exploring alternative evaluators or human verification would help strengthen the study’s methodological rigor.

- **Domain Generalization Gaps**
  The dataset’s current emphasis on general-purpose agent tasks limits its applicability to specialized, high-stakes domains. The absence of scenarios from critical areas such as finance, healthcare, and other regulated industries reduces the benchmark's value for evaluating agents intended for real-world, domain-specific deployments. Expanding into these professional verticals would substantially enhance both the practical relevance and technical robustness of the framework.

**Strengths Contributions:**

- Introduces **AGENTIF**, the first benchmark for evaluating LLMs’ instruction-following capabilities in realistic, complex agentic scenarios — a crucial yet underexplored area.
- Proposes a comprehensive constraint taxonomy, including unique **tool constraints**, and categorizes constraints into vanilla, conditional, and example types.
- Clearly distinguishes AGENTIF from prior benchmarks, which focus on short, synthetic instructions and lack complex, tool-based constraints, thereby effectively filling a critical gap in current evaluation standards.
- Provides a large, realistic dataset: **707 human-annotated instructions**, averaging 1,723 words and approximately 12 constraints each — representing a significant advance in scale and realism.
- The paper is **well-written, logically organized, and easy to follow**, with informative figures and tables that effectively support the findings.

---

> ### Author Rebuttal · Authors · 2025-07-31
>
> Thanks for your valuable comment!
>
> - **Regarding the concern about Limited Scope of Tool Constraints (Weakness #1)**
>
>     The tool constraints primarily cover basic requirements, as these are the most common scenarios among the agents we collected.
>
>     Thank you for the suggestion. There exist more complex tool use cases in real-world applications. We encourage the community to contribute additional agents. We believe our data construction pipeline can scale to these scenarios and support broader evaluation.
>
> - **Regarding the concern about Potential Evaluation Circularity (Weakness #2)**
>
>     Thanks for your feedback.
>     **We also use QwQ-32B as the evaluator and find its judgments to be highly consistent with those of GPT-4o (agreement scores exceeding 85%).** Specifically, we re-evaluate the top 4 LLMs, including o1-mini, GPT-4o, Qwen3-32B, and QwQ-32B using QwQ-32B. The table below shows the high aggrement rates between these two evaluators, which validates GPT-4o as the evaluator.
>
>     |  | QwQ-32B agreement with GPT4o (%) |
>     | --- | --- |
>     | o1-mini | 88.0 |
>     | GPT-4o | 87.2 |
>     | Qwen3-32B | 87.1 |
>     | QwQ-32B | 88.3 |
>
>     In addition, we conduct a manual evaluation and find a high agreement rate with GPT-4o’s assessments. Specifically, we sample 200 responses from the same set of models and manually evaluate whether each response satisfies each constraint. **The agreement rate between human and GPT-4o is 94%**. We believe it is sufficiently effective for automated evaluation.
>
>     Thank you for the feedback again. We will include these explanations in the paper.
>
> - **Regarding the concern about Domain Generalization Gaps (Weakness #3)**
>
>     Thank you for the suggestion.
>     Our benchmark includes a diverse set of agents across **10 domains such as legal, health, and enterprise**, as shown in the table below.
>
>     | Category | Application Domain | % of Dataset |
>     | --- | --- | --- |
>     | General | General Question Answering | 15.3 |
>     |  | Knowledge Reasoning | 8.5 |
>     |  | Information Retrieval and Search Optimization | 1.6 |
>     |  | Conversational AI Assistant | 1.6 |
>     |  | General Task Agent | 1.4 |
>     | Health | Mental Health Support | 10.9 |
>     |  | Health Advisory | 1.6 |
>     | Legal | Law Question Answering | 9.9 |
>     | Tourism | Audio Tour | 8.2 |
>     | Coding | Code Collaboration | 7.2 |
>     |  | Software Engineering | 1.6 |
>     |  | Web App Development | 1.4 |
>     |  | CLI-based Coding Assistant | 1.4 |
>     | Gaming | Game Design | 6.2 |
>     | Research | Research Assistant | 4.7 |
>     |  | News Intelligence | 3.1 |
>     |  | Research Writing | 1.6 |
>     | Enterprise | Enterprise Strategy And Meeting Intelligence | 3.5 |
>     |  | Enterprise Operations | 3.1 |
>     |  | Lead Generation | 1.6 |
>     |  | Market Strategy | 0.4 |
>     | Gaming | Game AI | 3.1 |
>     | Finance | Personal Finance Management | 2.3 |
>
>     We have collected as many diverse agents as possible for evaluation. We believe our data construction pipeline can scale to these professional scenarios with limited human efforts and encourage the open-source community to contribute more agents to further expand our evaluation. We will include a more detailed dataset distribution in the revised version.

---

### Official Review · Reviewer_Cfwh · 2025-07-03

**Rating:** 6
**Confidence:** 4

**Summary:**

The paper introduces AGENTIF benchmark for evaluating LLMs in agentic instruction-following scenarios. In contrast to previous instruction following benchmarks which cover shorter general instructions, AGENTIF covers realistic, long (>1000 words on average) instructions with complex constraints (11.9 constraints on average) for agentic systems (across 50 agentic real-world application examples). The paper tests a variety of popular language models (from small open source LLMs to flagship API-based models) on the benchmark and presents curious findings showcasing that current models underperform in long instruction following (with performance completely breaking down for >6000 word instructions) and constraint interpretation especially in cases where meta constraints on constraint selection are present.

**Dataset Code Accessibility:**

Yes

**Dataset Code Comments:**

the code is provided

**Ethical Considerations:**

No, there are no or only very minor ethics concerns

**Final Justification:**

This is a valuable contribution to the community and a paper with lasting impact, especially now that it covers most production-grade models. I am happy to raise my score to 6.

**Limitations Weaknesses:**

Weaknesses:
- Claude 3.7 and Claude 4 are missing, these are arguably the most capable agentic LLMs to date and it would be valuable to include them in evaluation
- More exploration of instructions with in-context learning examples would be helpful
- The paper mentions 50 real world agentic applications but does not provide enough specifics. Could you clarify what are the 50 applications used for benchmark construction?
- The benchmark might be difficult to scale because evaluation methods are sample-specific requiring custom LLM prompts and/or code for each instruction in the benchmark (as mentioned in Figure 3 and in the example “For example, when evaluating the constraint “The abstract should be no less than 100 words”, the method first uses an LLM to extract the abstract section, followed by code verification to assess word count compliance.”)
- The failure mode analysis could benefit from more hypotheses on failure root cause

**Strengths Contributions:**

Very timely, important, and comprehensive benchmark

Strengths:
- AGENTIF is a comprehensive benchmark which addresses an important need of benchmarking realistic agentic instruction following for since real world applications indeed require very long detailed instructions with potentially a large number of tools (and best practices by e.g. Anthropic recommend making the tool prompts very detailed too). Overall, AGENTIF is important and timely.
- Across 707 human-annotated instructions, AGENTIF covers three types of constraints in instructions: formatting, semantic, and tool constraints across different presentation types from plain text to examples to conditional constraints. These are indeed realistic and common constraints in production agentic systems. In total, the benchmark includes 8415 high quality constraints.
- The benchmark extensively evaluates 16 popular LLMs
- The benchmark evaluates instruction following at constraint level (CSR) and full instruction/system prompt level (ISR). The paper presents very interesting findings:
    - all models demonstrate suboptimal performance on long instructions (best CSR of 59.8%, best ISR 27.2%, ISR further drops to almost 0 for instructions longer than 6000 words) demonstrating an overlooked gap in agentic LLM capabilities.
    - As much as 25% of real world instructions use meta constraints of which the most popular one is constraint selection — asking the LLM to ignore some constraints and follow the others. On such constraint selection meta constraints, the models fail at an remarkable rate of 91.4% highlighting another mismatch between the use of agentic LLMs in production and their training objectives in development
    - The paper also reports (intuitively) that reasoning helps at instruction following
- The paper also presents detailed failure mode and error analysis

---

> ### Author Rebuttal · Authors · 2025-07-31
>
> Thanks for your valuable comments!
>
> - **Regarding the Performance of Claude Models (Weakness #1)**
>
>     We add the evaluation of **Claude 3.7 Sonnet** and **Claude 4 Sonnet**.  We find that Claude performs particularly well in tool constraints, which aligns with its advantage as an advanced agentic LLM. We will add these results to the paper.
>
>     | Models | Vanilla | condition | example | formatting | semantic | tool | ISR | CSR |
>     | --- | --- | --- | --- | --- | --- | --- | --- | --- |
>     | claude-sonnet-4 | 59.1 | 44.3 | 83.6 | 62.2 | 62.2 | 45.5 | 21.5 | 60.1 |
>     | O1-mini | 59.8 | 37.5 | 80.8 | 66.1 | 59.1 | 43.2 | 26.9 | 59.8 |
>     | claude-3-7-sonnet | 60.9 | 38.9 | 69.2 | 60.1 | 61.3 | 50.9 | 23.3 | 59.5 |
>
> - **Regarding the Analysis of Example Constraints (Weakness #2)**
>
>     Thank you for the suggestion. We analyze errors related to instructions with in-context learning examples (example constraint). Specifically, **we manually add the corresponding explicit constraint for each example constraint.** We observe three patterns: *still*, where the performance remains unchanged; *correct → wrong*, indicating the model fails to follow the constraint after adding corresponding explicit constraints; and *wrong → correct*, where the model follows the constraints after adding corresponding explicit constraints.
>
>     As shown in the table below, adding explicit constraints improves the model’s instruction-following ability. This suggests that **the primary challenge for current models lies in inferring implicit constraints from examples, rather than executing explicit ones.**
>
>     We will include a detailed description of this analysis in the revised version of the paper.
>
>     |  | still (%) | correct → wrong (%) | wrong → correct (%) |
>     | --- | --- | --- | --- |
>     | GPT-4o | 91.2 | 3.9 | 2.1 |
>     | QwQ-32B | 88.9 | 4.6 | 5.9 |
>     | Claude-3-5-Sonnet | 72.5 | 8.4 | 16.9 |
>
> - **Regarding concerns about dataset diversity (Weakness #3)**
>
>     Apology for any confusion. Our benchmark covers about **10 categories and 23 agent application scenarios**, as shown in the table below.  We will add further clarification in the revisions.
>
>     | Category | Application Domain | % of Dataset |
>     | --- | --- | --- |
>     | General | General Question Answering | 15.3 |
>     |  | Knowledge Reasoning | 8.5 |
>     |  | Information Retrieval and Search Optimization | 1.6 |
>     |  | Conversational AI Assistant | 1.6 |
>     |  | General Task Agent | 1.4 |
>     | Health | Mental Health Support | 10.9 |
>     |  | Health Advisory | 1.6 |
>     | Legal | Law Question Answering | 9.9 |
>     | Tourism | Audio Tour | 8.2 |
>     | Coding | Code Collaboration | 7.2 |
>     |  | Software Engineering | 1.6 |
>     |  | Web App Development | 1.4 |
>     |  | CLI-based Coding Assistant | 1.4 |
>     | Gaming | Game Design | 6.2 |
>     | Research | Research Assistant | 4.7 |
>     |  | News Intelligence | 3.1 |
>     |  | Research Writing | 1.6 |
>     | Enterprise | Enterprise Strategy And Meeting Intelligence | 3.5 |
>     |  | Enterprise Operations | 3.1 |
>     |  | Lead Generation | 1.6 |
>     |  | Market Strategy | 0.4 |
>     | Gaming | Game AI | 3.1 |
>     | Finance | Personal Finance Management | 2.3 |
>
>  - **Regarding concerns about dataset scalability（Weakness #4）**
>
>     Although the evaluation methods are sample-specific, they are also automatically generated. **Our automated data construction pipeline (Section 3.2)** covers both constraint extraction and **the generation of corresponding verification methods**. We manually verify each step to ensure the accuracy of the benchmark evaluation.
>
>     We find that about 90% of the extracted constraints and 80% of the generated verification methods are correct. Therefore, we believe our pipeline can scale to a wider range of scenarios with limited human effort. We encourage the community to contribute more open-source agents and further expand our evaluation.
>
> - **Regarding Failure Root Causes（weakness #5）**
>
>     Thanks for your suggestion.
>     We conduct **a detailed error analysis (Section 4.3) to explore possible failure reasons** and provide potential explanations. However, identifying the root causes requires deeper investigation, such as mechanistic interpretability (SAEs or neuron-level analysis) or conducting controlled training experiments. As these require substantial effort, we consider them out of scope for this work and leave them in future work.

---

### Official Review · Reviewer_A2Xs · 2025-07-07

**Rating:** 5
**Confidence:** 3

**Summary:**

This paper proposes AGENTIF to assess LLMs' adherence to user instructions. Compared to IFEval, AGENTIF features more realistic, longer, and complex instructions that better evaluate real-world compliance. The benchmark includes 707 human-annotated instructions covering formatting, semantic, and tool-based constraints, evaluated through code verification, LLM validation, and hybrid methods. Experiments reveal significant shortcomings in current LLMs: struggles with complex constraints/tool specifications, performance degradation with longer instructions, and emerging failures on meta-constraints.

**Additional Feedback:**

Questions:

1. Regarding the practical demands cited in the Introduction (Web Agents, Education Agents, GUI Agents, PPT Agents), were corresponding tasks incorporated into the subsequent instruction set and dataset?
2. The code verification appears exclusively Python-based. Could the authors consider extending compatibility to other programming languages to enhance benchmark generality?
3. While constraint count correlates with complexity, it is not always the case. Are there any alternative complexity metrics beyond quantity?



Minor:

Qwen3 [26] should be a thinking model, while the authors categorize it as a non-thinking model.

**Dataset Code Accessibility:**

Yes

**Dataset Code Comments:**

The dataset is publicly accessible via HuggingFace, with complementary evaluation code available in the GitHub repository. Following the repository's instructions, users can evaluate their models by simply changing the API key.

**Ethical Considerations:**

No, there are no or only very minor ethics concerns

**Final Justification:**

The responses substantively address my concerns, particularly given the authors' commitment to long-term maintenance of AGENTIF. This benchmark will establish critical standards for evaluating instruction-following capabilities.

**Limitations Weaknesses:**

This work faces key scalability limitations: the dataset construction requires manual annotation and verification, lacking an automated pipeline for generating high-quality instructions. Furthermore, while benchmark tasks are labeled "realistic," they derive primarily from GitHub agentic applications—predominantly coding scenarios. This contrasts with the industrial applications referenced in the introduction (e.g., industrial control systems or fine-grained manipulation tasks), which could better reflect LLMs' instruction-following capabilities across diverse sectors.

**Strengths Contributions:**

This well-structured paper presents a clear logical flow. AGENTIF addresses a critical gap in existing benchmarks by designing realistic, complex instructions that better reflect practical scenarios. The benchmark employs a robust evaluation triad—code-based checks, LLM validation, and hybrid methods—to assess instruction adherence. Its comprehensive instruction set features extended length, diverse constraint types (formatting, semantic, tool), and varied constraint presentations. Crucially, human verification was applied to all LLM-assisted content generation.



Experimental results validate intuitive expectations: comparative analysis of thinking/non-thinking  models using ISR and CSR metrics reveals consistent performance patterns. The authors provide valuable insights into failure modes across settings, supported by well-reasoned hypotheses and improvement pathways.



I believe this benchmark establishes a foundational platform for advancing LLMs' instruction-following capabilities and enhancing LLMs' controllability.

---

> ### Author Rebuttal · Authors · 2025-07-31
>
> Thanks for your valuable comments!
>
> - **Regarding concerns about Dataset Scalability（Weakness #1）**
>
>     We also think the scalability of data construction is important. Therefore, we **develop an automated data construction pipeline (Section 3.2)** to extract constraints and generate corresponding verification methods.  However, there is still noise in these steps, and as a benchmark, manual check and validation are necessary. Therefore, we conduct manual verification in the data construction process to ensure the quality of our benchmark.
>
>     We believe our data construction pipeline is scalable to a larger number of instances with limited human effort. However, due to the limited availability of agent system prompts, we focus solely on constructing a high-quality benchmark. In the future, we plan to adopt a similar approach to collect training data.
>
> -  **Regarding concerns about Dataset Diversity（Weakness #2，Feedback #1）**
>
>     Apology for any confusion. Our benchmark covers about **23 agent application scenarios**, such as coding, research, gaming, and enterprise, as shown in the table below. We will add further clarification in the revisions. We also welcome community contributions of additional agent system prompts to expand our evaluation.
>
>     | Category | Application Domain | % of Dataset |
>     | --- | --- | --- |
>     | General | General Question Answering | 15.3 |
>     |  | Knowledge Reasoning | 8.5 |
>     |  | Information Retrieval and Search Optimization | 1.6 |
>     |  | Conversational AI Assistant | 1.6 |
>     |  | General Task Agent | 1.4 |
>     | Health | Mental Health Support | 10.9 |
>     |  | Health Advisory | 1.6 |
>     | Legal | Law Question Answering | 9.9 |
>     | Tourism | Audio Tour | 8.2 |
>     | Coding | Code Collaboration | 7.2 |
>     |  | Software Engineering | 1.6 |
>     |  | Web App Development | 1.4 |
>     |  | CLI-based Coding Assistant | 1.4 |
>     | Gaming | Game Design | 6.2 |
>     | Research | Research Assistant | 4.7 |
>     |  | News Intelligence | 3.1 |
>     |  | Research Writing | 1.6 |
>     | Enterprise | Enterprise Strategy And Meeting Intelligence | 3.5 |
>     |  | Enterprise Operations | 3.1 |
>     |  | Lead Generation | 1.6 |
>     |  | Market Strategy | 0.4 |
>     | Gaming | Game AI | 3.1 |
>     | Finance | Personal Finance Management | 2.3 |
>
> - **Regarding Code Verification（Feedback #2）**
>
>    Thank you for the suggestion. We use Python as it is a widely used and convenient scripting language. Our verification method is language-agnostic, and we encourage the community to adapt the Python code to other programming languages to enhance the generality of our evaluation.
>
> - **Regarding Constraint Quantity Analysis（Feedback #3）**
>
>    Thank you for the insightful observation. As shown in Section 4.4 and Figure 5(b), performance generally declines with more constraints and longer instructions, indicating their correlation with complexity. We consider it a suitable and convenient proxy for instruction complexity.
>
>    Other factors, such as constraint type, may also influence complexity. We leave more in-depth analysis to future work.
>
> - **Regarding the Typo (Minor)**
>
>    Thank you for the careful observation. We acknowledge the mislabeling and have corrected it in our revised version. This correction does not affect any of the main conclusions of our paper.

---

### Comment · Area_Chair_kvE8 · 2025-08-06

Dear Reviewers,

Thank you for sharing your valuable insights and expertise, which have played an important role in the review process.

In response to the initial feedback, the authors have submitted a detailed rebuttal addressing the comments raised by the reviewers.

I would appreciate it if you could carefully review their response and consider how it may affect your initial evaluation.

Please feel free to share your updated thoughts or any additional comments after reviewing the rebuttal.

Thank you again for your time and contributions.

---

### Note · Authors · 2025-08-12

We thank the reviewers and AC for their time and valuable feedback. We believe we have adequately responded to all concerns and would like to briefly highlight our contributions and key discussions.
### **Contributions of Our Paper**
- We propose AgentIF, a benchmark for evaluating instruction following in realistic agentic scenarios with long, detailed instructions and complex constraints. We believe AgentIF is valuable to the community for building robust agentic applications.
- We conduct comprehensive evaluations using AgentIF and draw several insights. Our results show that even the best-performing LLM follows fewer than 30% of instructions perfectly, demonstrating the difficulty and necessity of AgentIF.
- We conduct further analyses, including error analysis, instruction length, and meta instructions, which reveal specific limitations of existing LLMs and potential directions for future optimization.
### **Key Discussion 1: Scenario Diversity (Reviewers A2Xs, Cfwh, P5jt)**
Our benchmark covers **23 agentic application scenarios across 9 categories:** General, Legal, Health, Coding, Finance, Research, Enterprise, Tourism, and Gaming. We provide a detailed table in our response and will add further clarification in the revisions.
### **Key Discussion 2: Dataset Scalability (Reviewers A2Xs, Cfwh)**
We **develop an automated data construction pipeline (Section 3.2)** that covers both **constraint extraction and verification method generation**. We further conduct manual verification to ensure the benchmark quality. We find that about 90% of the extracted constraints and 80% of the generated verification methods are correct. Therefore, we believe our pipeline can scale to a wider range of scenarios with limited human effort.
### **Key Discussion 3: Reliability in LLM Evaluation (Reviewers P5jt, uQoP)**
Our evaluation integrates code-based and LLM-based evaluation methods, leveraging the proven effectiveness of LLM evaluation across diverse domains.

As suggested by the reviewers, we manually verify the reliability of GPT-4o's evaluations by randomly sampling 200 responses for constraint satisfaction. **The agreement between GPT-4o and human judgments reached 94%**, confirming GPT-4o’s adequacy for automated evaluation.

For potential evaluation circularity, **we use QwQ-32B as the evaluator and find its judgments to be highly consistent with those of GPT-4o (agreement rates exceeding 85%),** further supporting the reliability of LLM evaluation.

---

### Decision · Program_Chairs · 2025-09-18

**Decision:**

Accept (spotlight)

**Comment:**

This paper introduces AGENTIF, a comprehensive benchmark for evaluating LLM instruction-following ability in agentic scenarios. I recommend acceptance of this submission for the following reasons:

* Novel and realistic benchmark: AGENTIF comprises 707 human-annotated instructions that effectively address a critical gap in current evaluation standards. Unlike prior benchmarks that rely on short, synthetic instructions, AGENTIF features extended-length instructions with complex, tool-based constraints that better reflect real-world agentic scenarios.

* Robust evaluation methodology: The benchmark proposes a robust evaluation protocol combining code-based checks, LLM validation, and hybrid methods to ensure comprehensive assessment of instruction adherence.

* Comprehensive empirical analysis: The authors evaluate 16 popular LLMs using AGENTIF, providing valuable insights into current model capabilities. The detailed failure mode and error analysis offers actionable insights for future model development.

I think the paper makes a nice contribution that the community will find valuable. However, I encourage the authors to carefully address reviewer comments in the camera-ready version.